# Cardiovascular risk factors among nurses: A global systematic review and meta-analysis

Saghar Khani[1], Sima Rafiei[2], Ahmad Ghashghaee[3]*, Maryam Masoumi[4], Srva Rezaee[5], Golnaz Kheradkhah[6], Bahare Abdollahi[1]

1 Student Research Committee, School of Medicine, Iran University of Medical Sciences, Tehran, Iran, 2 Social Determinants of Health Research Center, Research Institute for Prevention of Non-Communicable Diseases, Qazvin University of Medical Sciences, Qazvin, Iran, 3 The School of Medicine, Dentistry & Nursing, University of Glasgow, Glasgow, United Kingdom, 4 Clinical Research and Development Center, Qom University of Medical Sciences, Qom, Iran, 5 Instructor Clinical Care Research Center, Research Institute for Health Development, Faculty of Nursing and Midwifery, Kurdistan University of Medical Sciences, Sanandaj, Iran, 6 Student Research Committee, School of Nursing and Midwifery, Iran University of Medical Sciences, Tehran, Iran

* Ahmad.ghashghaee1996@gmail.com

## Abstract

**Data Availability Statement:** All relevant data are within the paper and its Supporting Information files.

### Background

Nurses as the largest group of health workers have a very stressful job which can cause number of diseases specially increase cardiovascular risk factors. This study aims to investigate the overall epidemiology of cardiovascular disease (CVD) risk factors among nurses.

### Method

We searched all four main databases such as Scopus, PubMed, Embase and Web of Sciences from the beginning of 2000 to March 2022 with appropriate Mesh Terms. We also searched Google scholar. Then we applied inclusion and exclusion criteria and after selection the studies the Newcastle-Ottawa Scale (NOS) was used to assess the methodological quality of included studies. Comprehensive Meta-analysis and R software was used for analysis.

### Results

Finally, 22 articles with a total number of 117922 nurses were included. Among all risk factors, sedentary lifestyle and lack of regular physical activity with a prevalence of 46.3% (CI 95%, 26.6–67.2) was regarded as the main prevalent risk factor among nurses. The mean systolic blood pressure (SBP) measured in the study population was 121.31 (CI 95%, 114.73–127.90) and the mean diastolic blood pressure (DBP) was 78.08 (CI 95%, 74.90–81.25). Also family history of cardiovascular disease (41.9%; 95% IC: 29.8–55.1%), being overweight (33.3%; 95% IC: 24.7–43.2%), and alcohol consumption (24.6%; 95% IC: 16.4–35.2%) was found among the participants.

**Funding:** The authors received no specific funding for this work. The funders had no role in study design, data collection and analysis, decision to publish, or preparation of the manuscript.

**Competing interests:** The authors have declared that no competing interests exist.

## Conclusion

**S**tudy results revealed that sedentary lifestyle was the main prevalent CVD risk factor among nurses followed by family history of cardiovascular disease, being overweight and alcohol consumption. Furthermore, among nurses with shift works almost all risk factors got higher score representing the worse condition in comparison with day workers' nurses. This study enables learning the associated risk factors of CVD among nurses to facilitate interventional programs with a view to reduce the exposure of nursing staff particularly those who work in shifts to cardiovascular risk factors.

## 1. What was already known?

In general, many studies have emphasized the impact of the nursing profession on the incidence of some cardiovascular patients. Also, different shifts of nurses can have a double effect.

## 2. What are the new findings?

In this study, the mean for sedentary lifestyle was reported to be 46.3% which represented the most prevalent risk factor for cardiovascular risk factors among study population.

## 3. What is their significance?

This study enables learning the associated risk factors of CVD among nurses to facilitate interventional programs with a view to reduce the exposure of nursing staff particularly those who work in shifts to cardiovascular risk factors. This information can comprise essential tools for health human resource management contributing to advance nursing.

## Introduction

Cardiovascular disease is the main cause of mortality and severe disabilities worldwide. In 2019, the disease prevalence increased to 523 million and its mortality is expected to constitute 31.7% of global deaths by 2030 [1]. Increased prevalence of cardiovascular risk factors such as obesity, diabetes, high cholesterol, and hypertension has caused a considerable increase in mortality rates among CVD (Cardiovascular Disease) patients [2]. In addition, some of the modifiable risk factors such as smoking, poor diet, alcohol consumption, and sedentary lifestyle are attributed to the majority of CVD cases and deaths [3, 4]. The role of psychological factors including work stress and job strain has also been investigated in different literatures as potential risk factors for coronary heart diseases [5].

Nurses as the largest group of health care providers should constantly promote their health status due to an intertwining connection between the quality of services rendered by them and their health performance [6–8]. Furthermore, nursing is a stressful job which necessitates nurses to work in long, variable shifts, and expose to a variety of work-related hazards, stress, and burnout leading to various health complications such as cardiovascular diseases [9]. Researchers have investigated that rotating shift workers are associated with a higher prevalence of coronary risk factors including smoking, hypertension, increased serum cholesterol, uric acid, and glucose levels together with depression, obesity, sleep disorders and behavioral changes [10, 11].

On the other hand, heart diseases not only impose several direct and indirect costs to the health system, but also they can result in job burnout, absenteeism, loss of productivity, or job instability among healthcare providers [12]. Given the key role of nurses in the health delivery system, their mental disorders and consequently physical illnesses might have devastating effects on the quality of provided services to patients [13]. Therefore, to develop effective strategies for the prevention of cardiovascular diseases among nurses, it is required to determine and classify associated risk factors, and take necessary actions to prevent complications in at-risk nursing staff [14].

Different literatures on cardiovascular risk factors among nursing staff have shown that a considerable number of nurses are in face of an increased level of risk factors particularly overweight, lack of regular exercise, sleep disorders, smoking and poor diet [8–14]. In a research conducted by Miller et al. findings revealed that 96% of obese nurses mentioned their obesity as a risk factor of heart disease, while nearly 30% of them were not aware of their diabetes, and about 90% were uninformed of having hyperlipidemia [8]. Even though nurses are continuously monitoring the health condition of patients, they rarely pay necessary attention to their health issues. Due to an increasing rate of coronary vascular disease and related deaths worldwide and lack of sufficient evidence dealing with the epidemiology of CVDs and its risk factors among nurses we conducted a systematic review and meta-analysis to describe the overall epidemiology of CVD risk factors in this group of key health care providers globally.

## Materials and methods

### Study design and registration

A systematic review was conducted based on the Preferred Reporting Items for Systematic reviews and Meta-Analyses guidelines (PRISMA) [15]. Also the protocol of this SLR has been registered on PROSPRO with code CRD42022329794.

### Search strategies for identification of studies

A systematic search of SCOPUS, Web of Science, Google Scholar, EMBASE and PubMed was done from the beginning of 2000 to the end of March 2022 through the following search terms including ((Heart Disease Risk Factors[Title]) OR (Heart Disease Risk Factor[Title]) OR (Risk Factors for Heart Disease[Title]) OR (Risk Factor for Heart Disease[Title]) OR (Cardiovascular Risk Factors[Title]) OR (Cardiovascular Risk Factor[Title]) OR (Risk Factors for Cardiovascular Disease[Title]) OR (Cardiovascular Risk Score[Title]) OR (Cardiovascular Risk Scores[Title]) OR (Cardiovascular Risk[Title]) OR (Cardiovascular Risks[Title]) OR (Residual Cardiovascular Risk[Title]) OR (Residual Cardiovascular Risks[Title]) OR (Cardio metabolic Risk Factor[Title]) OR (metabolic Risk Factor[Title]) OR (metabolic Risk Factors[Title]) OR (Cardio metabolic Risk Factors[Title]) OR (Coronary Disease[Title]) OR (Coronary Heart Disease[Title]) OR (Coronary Heart Diseases[Title]) OR (Heart Diseases[Title]) OR (Heart Disease[Title]) OR (Cardiac Diseases[Title]) OR (Cardiac Disease[Title]) OR (Cardiac Disorders[Title]) OR (Cardiac Disorder[Title]) OR (Heart Disorders[Title]) OR (Heart Disorder [Title]) OR (Cardiovascular Diseases[Title]) OR (Cardiovascular Disease[Title]) OR (Hypertension[Title]) OR (High Blood Pressure[Title]) OR (High Blood Pressures[Title]) OR (Vascular Diseases[Title]) OR (Vascular Disease[Title])) AND ((Nurse[Title]) OR (Nurses[Title]) OR (Nursing Personnel[Title]) OR (Registered Nurses[Title]) OR (Registered Nurse[Title]) OR (Hospital Personnel[Title]) OR (hospital staff [Title])).

## Data collection

Following the search of electronic databases, 334 articles were identified. After importing the documents to EndNote software, duplicates were removed and finally resulted in 221 articles; of which 76 articles were published in Pub Med, 91 in SCOPUS, 23 in Web of Science and 31 articles were retrieved from EMBASE. Then, title/ abstracts were screened by two independent members of the research team in order to check data relevancy. Studies which reported quantitative data on CVD prevalence among nurses and its risk factors including (SBP(Systolic Blood Pressure) $\geq$ 140mmHg, DBP(Diastolic Blood Pressure) $\geq$ 90mmHg, HDL(High-Density Lipoprotein) <40, LDL(Low-Density Lipoprotein) >100, TC(Total Cholesterol) $\geq$200, Obesity (BMI$\geq$30), smoking, physical inactivity, alcohol consumption 3 or 4/wk, high blood sugar, having the history of CVD, or family history of CVD) were included for further consideration. References of included articles were examined and conference abstracts were also searched to be included as additional references. After applying inclusion/exclusion criteria, 22 studies were found to be included in the research (Fig 1).

## Inclusion and exclusion criteria

Both longitudinal and cross-sectional studies published in English from 2000 to March 2020 with quantitative data on CVD prevalence among nurses and its risk factors were included in the review. On the other hand, other types of scientific evidence including expert opinion, randomized controlled trial, commentaries, letter to editor, thesis, brief report, case-control, case-series, book chapter, and editorial, were not included in the research. In addition, studies with insufficient data on research questions, and those emphasizing on diagnosis or therapeutic approaches and medication therapies were excluded from review. Other inclusion criteria were studies reporting nurses who were suffering from cardiovascular disease or other outcomes including coronary heart disease, hypertensive disease, rheumatic heart disease, cerebral vascular disease, and other cardiovascular diseases which constitute a condition affecting the heart and blood vessels associated with a make-up of fatty substances, cholesterol and deposits within an artery. The diseases generally occur when these buildups cause the arteries to narrow and decrease blood flow to the heart and consequently lead to the shortness of breath, chest pain, or even heart attack.

## Data extraction

Data extraction was done by two independent investigators using a developed form encompassing name of author/ authors, date of publication, research setting, study design, study objective in terms of reporting SBP $\geq$ 140mmHg, DBP $\geq$ 90mmHg, HDL<40, LDL>100, TC$\geq$200, obesity (BMI$\geq$30), smoking, physical inactivity, alcohol consumption 3 or 4/wk, high blood sugar, history of CVD, and family history of CVD. Kappa index was used to test the concordance between investigators. As the calculated value was 0.78, it showed a good concordance. In terms of any disagreement, a third reviewer was asked to resolve the issue.

## Quality assessment

The Newcastle-Ottawa Scale (NOS) was used to assess the methodological quality of included studies. The NOS consists of eight questions in three main sections including ascertainment of exposure/ outcome, selection of study groups, and their comparability. Unreported and reported items were scored 0 and 1 respectively, and total score of each paper was estimated by considering the sum of scores allocated to the reported items. An article with score below four was considered as a low level of quality [16].

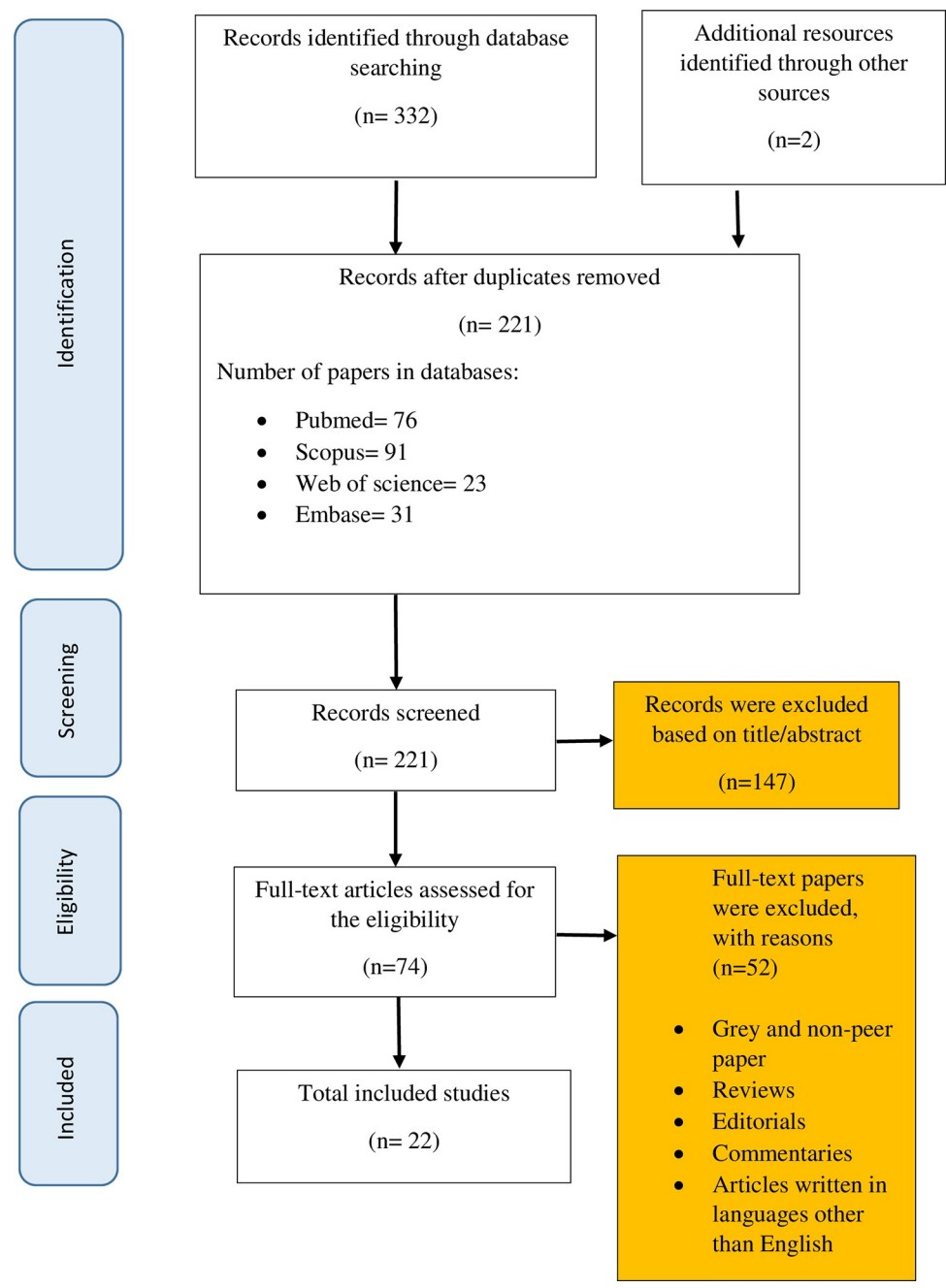

**Fig 1. Flow diagram of our review process (PRISMA).**

## Data analysis

The standardized mean difference effect size for the mean score and the prevalence of cardiovascular risk factors was measured. To estimate the mean and variability of effect size across studies random-effect analysis was used and the results were reported on a forest plot at a 95% confidence interval. In addition, the heterogeneity test ($I^2$) was done to determine effect size homogeneity between included studies. In case of heterogeneity in the areas of research setting, shift works subgroup analysis was performed. To analyze data, R software was applied.

**Table 1. The characteristics of studies.**

| First Author, Year, References | Year | Total sample | shift work | Total Male | Total Female | Country | Continent | WHO Regional classification |
|---|---|---|---|---|---|---|---|---|
| Solymanzadeh et al., 2021 [17] | 2022 | 120 | Shift workers | 26 | 34 | Iran | Asia | EMRO |
| Buremoh et al., 2020 [18] | 2020 | 196 | Shift workers | NG | NG | Nigeria | Africa | AFRO |
| Fair et al., 2009 [19] | 2009 | 1345 | Day workers | 51 | 1294 | United States | America | AMRO |
| Burns et al., 2010 [13] | 2010 | 103 | Shift workers | 93 | 10 | United States | America | AMRO |
| Silva et al., 2016 [20] | 2017 | 20 | Shift workers | 2 | 18 | Brazil | America | AMRO |
| Faruque et al., 2021 [21] | 2021 | 938 | Shift workers | 114 | 824 | Bangladesh | Asia | SEARO |
| Gallagher et al., 2018 [22] | 2017 | 5041 | Shift workers | 620 | 4421 | Australia | Australia | WPRO |
| HA et al., 2005 [23] | 2004 | 226 | Shift workers | 0 | 226 | Korea | Asia | SEARO |
| Zhao et al., 2019 [24] | 2021 | 84697 | Day workers | 0 | 84697 | China | Asia | WPRO |
| Yan and Xie, 2022 [25] | 2021 | 1344 | Shift workers | 0 | 1344 | China | Asia | WPRO |
| Martínez-Gurrión et al., 2014 [26] | 2014 | 195 | Shift workers | 9 | 186 | Mexico | America | AMRO |
| Monakali, 2018 [27] | 2018 | 203 | Shift workers | 24 | 179 | South Africa | Africa | AFRO |
| Nobahar et al., 2015 [28] | 2015 | 56 | Shift workers | 6 | 50 | Iran | Asia | EMRO |
| Donkor et al., 2020 [29] | 2016 | 95 | Day workers | 32 | 63 | Liberia | Africa | AFRO |
| Riese et al., 2000 [6] | 2000 | 165 | Day workers | NG | NG | Netherlands | Europe | EURO |
| Hansen et al., 2016 [30] | 2022 | 20701 | Day workers | NG | NG | Denmark | Europe | EURO |
| Sahebi et al., 2010 [31] | 2010 | 542 | Shift workers | NG | NG | Iran | Asia | EMRO |
| Griep et al., 2015 [32] | 2015 | 388 | Day workers | 73 | 315 | Brazil | America | AMRO |
| Saberinia et al., 2020 [33] | 2018 | 250 | Shift workers | NG | NG | Iran | Asia | EMRO |
| Miller et al., 2008 [8] | 2007 | 749 | Shift workers | NG | NG | United States | America | AMRO |
| Bahadar Khan et al., 2012 [7] | 2012 | 165 | Shift workers | 11 | 154 | Pakistan | Asia | EMRO |
| Jahromi et al., 2017 [34] | 2017 | 263 | Day workers | 53 | 210 | Iran | Asia | EMRO |

# Results

This systematic review was reported according to the Preferred Reporting Items for Systematic Review and Meta-analysis (PRISMA). Finally, 22 articles with a total number of 117922 nurses were included in this systematic review. The characteristics of included studies are summarized in Table 1.

## Meta-analysis based on behavioral risk factors

Among all risk factors, sedentary lifestyle and lack of regular physical activity with a prevalence of 46.3% (CI 95%, 26.6–67.2) was regarded as the main prevalent risk factor among nurses. Furthermore, other prevalent risk factors were family history of cardiovascular disease (41.9%; 95% IC: 29.8–55.1%), and being overweight (33.3%; 95% IC: 24.7–43.2%) followed by alcohol consumption (24.6%; 95% IC: 16.4–35.2%). However, smoking had the least prevalence reported among 3.9% (CI 95%, 1.6–9.3) of nurses (Table 2).

## Meta-analysis based on physical risk factors

The mean systolic blood pressure (SBP) measured in the study population was 121.31 (CI 95%, 114.73–127.90) and the mean diastolic blood pressure (DBP) was 78.08 (CI 95%, 74.90–81.25); nevertheless, 11.2% (CI 95%, 8–15.3) of nurses had SBP above 140 mm Hg and 13.5% (CI 95%, 9.6–18.7) of them had DBP above 90 mm Hg which considered as hypertension (Table 3). The mean Body mass index (BMI) among nurses was estimated at 25.51 (CI 95%, 24.18–26.84) while 33.3% (CI 95%, 24.7–43.2) of nurses with a BMI between 25 and 29.9 were over weighted and 15.3% (CI 95%, 10.7–23.3) of them with BMI> 30 were suffering from obesity (Table 3).

**Table 2. The prevalence of cardiovascular risk factors among nurses.**

| Cardiovascular risk factors | Effect size and 95% interval | | | Test of null (2-Tail) | |
|---|---|---|---|---|---|
| | Point estimate | Lower limit | Upper limit | Z-value | P-value |
| SBP ≥ 140mmHg | 0.112 | 0.080 | 0.153 | -11.104 | p < 0.001 |
| DBP ≥ 90mmHg | 0.135 | 0.096 | 0.187 | -9.432 | p < 0.001 |
| HDL<40 | 0.194 | 0.072 | 0.425 | -2.489 | 0.013 |
| LDL>100 | 0.123 | 0.022 | 0.467 | -2.103 | 0.035 |
| TC≥200 | 0.093 | 0.052 | 0.160 | -7.204 | p < 0.001 |
| BMI (25–29.9)overweight | 0.333 | 0.247 | 0.432 | -3.237 | 0.001 |
| Obese (BMI≥30) | 0.153 | 0.107 | 0.232 | -5.327 | 0.001 |
| Smoker | 0.039 | 0.016 | 0.093 | -6.740 | p < 0.001 |
| NO Physical inactivity | 0.463 | 0.266 | 0.672 | -0.338 | 0.035 |
| Alcohol 3 or 4/wk | 0.246 | 0.164 | 0.352 | -4.317 | p < 0.001 |
| High Blood Sugar | 0.039 | 0.016 | 0.094 | -6.749 | p < 0.001 |
| History of CVD | 0.026 | 0.008 | 0.081 | -5.960 | p < 0.001 |
| Family history of CVD | 0.419 | 0.298 | 0.551 | -1.208 | 0.027 |

## Meta-analysis based on biological risk factors

The mean fasting blood sugar was estimated at 85.52 (CI 95%, 79.80–91.23) while 3.9% of study participants (CI 95%, 1.6–9.3) were at risk of diabetes due to fasting or random blood sugar upper limit of normal. Analyzing nurses' lipid profiles showed that their mean total cholesterol, LDL and HDL were 174.24 (CI 95%, 161.63–186.86), 91.54 (CI 95%, 79.56–103.53) and 48.1 (CI 95%, 44.75–51.63), respectively. Furthermore, 9.3% of nurses (CI 95%, 5.2–16) had hypercholesterolemia (TC≥200); 12.3% of them (CI 95%, 2.2–46.7) had LDL≥ 100 and 19.4% (CI 95%, 7.2–42.5) had HDL<40, which were reported to be significantly associated with an increased risk of cardiovascular diseases. (Table 3).

## Meta-analysis based on family history of CVD or past medical history of the disease

Results of meta-analysis revealed that 41.9% (CI 95%, 29.8–55.1) of study population reported a family history of cardiovascular disease but only 2.6% of them (CI 95%, 0.8–8.1) reported a past history of the disease (Table 3).

## Meta-analysis based on work shifts

To better understand the effects of working hours on nurses, we divided all participants into two groups of day workers and shift workers. Study results showed that, among nurses with

**Table 3. The mean score of cardiovascular risk factors among nurses.**

| Indexes | Effect size and 95% confidence interval | | | | | Test of null (2-Tail) | |
|---|---|---|---|---|---|---|---|
| | Point estimate | Standard error | Variance | Lower limit | Upper limit | Z-value | P-value |
| BMI | 25.514 | 0.680 | 0.462 | 24.182 | 26.846 | 37.536 | p < 0.001 |
| TC | 174.246 | 6.437 | 41.432 | 161.630 | 186.862 | 27.070 | p < 0.001 |
| HDL | 48.198 | 1.755 | 3.081 | 44.758 | 51.639 | 27.457 | p < 0.001 |
| LDL | 91.546 | 6.115 | 37.393 | 79.561 | 103.531 | 14.971 | p < 0.001 |
| SBP | 125.317 | 3.359 | 11.283 | 114.733 | 127.900 | 36.117 | p < 0.001 |
| DBP | 78.083 | 1.619 | 2.622 | 74.909 | 81.257 | 48.219 | p < 0.001 |
| FBG (Fasting Blood Glucose) | 85.521 | 2.916 | 8.504 | 79.805 | 91.237 | 29.326 | p < 0.001 |

**Table 4. Meta-analysis based on work shifts.** Mean (SD).

| Indexes | Day workers | Shift workers | P-value |
|---|---|---|---|
| BMI | 24.1 (2.5) | 27.2 (3.2) | 0.02 |
| TC | 170.36 (10.2) | 176.47 (9.6) | 0.13 |
| HDL | 49.72 (6.5) | 47.24 (4.5) | 0.03 |
| LDL | 89.74 (7.6) | 94.36 (8.4) | p < 0.001 |
| SBP | 122.24 (9.6) | 132.69 (5.3) | 0.12 |
| DBP | 72.36 (2.5) | 82.21 (3.6) | 0.32 |
| FB | 84.96 (11.25) | 95.14 (9.1) | p < 0.001 |

shift works almost all risk factors got higher score representing the worse condition. To cite and example, the mean score of total cholesterol was estimated at 170.36 (10.2) among day workers' nurses, however, the score among shift workers was reported to be 176.47 (9.6) (Table 4).

## Publication bias

According to the analysis regarding the detection of publication bias (shown in Funnel plot in Fig 2), there was no publication bias and the Egger test results were below one (P = 0.68) (Fig 2).

## Discussion

In this study, among all risk factors, sedentary lifestyle and lack of regular physical activity with a prevalence of 46.3% (CI 95%, 26.6–67.2) was regarded as the main prevalent risk factor among nurses. Although the prevalence rate for physical inactivity was different in several researches (for example 86.9%, 72.6%, and 40.6% in Faruque et al., Samir et al. and Al-Hazzaa

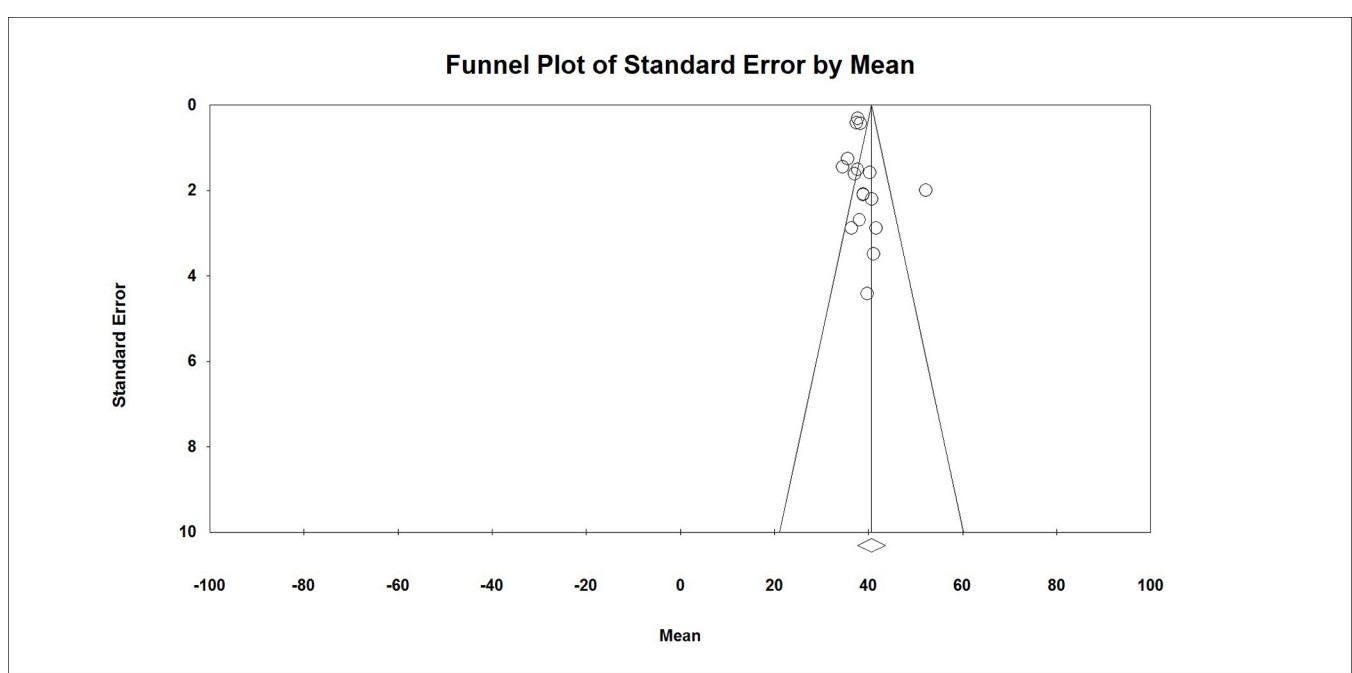

**Fig 2. Funnel plot of publication bias.**

studies respectively), most of them agreed on physical inactivity as the most significant cardio-vascular risk factor threatening the health status of nurses [21, 35, 36]. Correspondingly, in a study by AboSaad, physical inactivity was mentioned to be nearly two times more prevalent among healthcare personnel compared to the general population (55.9% compared to 29.8%) [37]. Circadian rhythm disturbances, lack of adequate time to get rest, work strain, fatigue and stress are regarded as the most common barriers to a regular physical activity among health workforce. Meanwhile, having regular exercise helps nurses remain the physical resilience required to overcome stressful situation and hold an adequate level of rigor in their job. It also helps them prevent serious physical disabilities such as heart diseases and other chronic condi-tions [38]. Furthermore, various studies have confirmed that an increased physical activity could be associated with decreased prevalence of CVD due to its contribution to reduced hypertension, obesity, LDL, and diabetes. In fact, exercise has protective effects to decrease the overall risk of CVD incident [39]. Therefore keeping regular physical activity besides having a healthy diet can play key role in avoiding CVD [40].

Furthermore, our study revealed that the second most prevalent risk factor among nurses was alcohol consumption. Several reports have confirmed the linking between alcohol con-sumption and increased prevalence of CVD. They also indicated the significant effect of a low alcohol consumption on the reduction of CVD risk [26-28]. Health care workers particularly nurses are greatly prone to hard work, stress, burnout, and sleep disorders which results in dif-ferent health disorders particularly cardiovascular disease [29]. There is evidence that 42.6% of health workforce in the US had alcohol use disorder due to several stressors encountering in the workplace resulting in burnout and occupational stress [26]. In a study 26.6% of nurses were reported to consume alcohol above the limit, which was considerably higher than that reported in the general population (17% versus 9%) [41]. Similarly, in Nigeria, the prevalence of binge drinking among nurses was higher than the rest of community (9.1% versus 1%) [28]. In coun-tries with limited resources, effective strategies should be considered for the prevention of heart disease through controlling potential risk factors particularly those related to behavioral habits [29, 30]. Thus, lifestyle modifications such as controlling alcohol drinking, stress, negative emo-tions, and sleep disorders can properly manage and prevent CVD among nurses [42].

Cigarette smoking is another risk factor which contributes to the high prevalence of CVD and has been responsible for nearly 140,000 annual deaths from the disease [32]. Based on our study findings, 3.9% of nurses were current smokers which was higher than that reported in the studies by Faruque; while was less than that reported by Ohida's study [21, 43]. As it is obvious from different study results, smoking had the lowest prevalence among other risk fac-tors of CVD. For example, Khan et al. reported that only 0.6% of nurses smoked, and Fair et al. estimated the prevalence of smoking among nurses at 3.6% [7, 19]. Despite the low prevalence of this risk factor among nurses with CVD, having adequate knowledge about smoking side effects and being in a position of role modeling smoking cessation seem to act as fundamental barriers for nurses to avoid smoking [35, 36].

The mean SBP measured in the study population was 121.31 and the mean DBP was 78.08; nevertheless, 11.2% of nurses had SBP above 140 mm Hg and 13.5% of them had DBP above 90 mm Hg which considered as hypertension. Our findings also revealed that approximately 47% of coronary heart diseases are attributable to high blood pressure. However, the preva-lence of hypertension vary significantly worldwide from 1.5% in Fair's study to 20% in Galla-gher's [22]. These differences might be due to the variations in the demographic characteristics of study population (including age, gender, family history, work shift, etc.) as well as differ-ences in design, analysis and interpretation of study findings. Evidence has affirmed that nurses working in shifts had a considerable increased level of blood pressure and an aug-mented cardiovascular risk [44]. A similar study showed that estimated systolic blood pressure

was at 122.04 mmHg in shift workers and 115.62mmHg in day workers. Additionally, the diastolic blood pressure was reported to be 81.9 mmHg in shift workers and 79.8 mmHg in day workers. Accordingly, the prevalence of hypertension was almost four times higher among shift workers (20% in shift workers versus 4% in day workers) [45]. Similarly, in another research by Abedini et al. the prevalence of hypertension in shift workers was almost twice the prevalence in day workers. In this study, the mean SBP in shift workers was 141.05 mmHg, which was higher than day workers estimated at 22.45 mmHg; while, the mean DBP of shift workers was reported to be slightly lower than day workers (9.53 mmHg versus 10.46 mmHg) [46]. Other studies also affirmed the findings and agreed on higher prevalence of SBP and DBP in shift workers compared to day workers [47]. Thus, it is recommended that health promotion education, screening programs, and continuous care of nurses should be included in the health and wellbeing programs of nursing staff to control their high blood pressure as a main risk factor for CVD [46].

According to study analysis, the mean Body Mass Index (BMI) among nurses was estimated at 25.51 while 33.3% of them with a BMI between 25 and 29.9 were over weighted and 15.3% with BMI> 30 were suffering from obesity. Relatively, Griep et al. found that a 10 kg higher body weight is related to a 3.0 mm Hg higher systolic and 2.3 mm higher diastolic blood pressure which consequently results in a significant 12% increase in coronary heart disease and 24% augmented risk of stroke [47]. The high rate of obesity signifies an inappropriate lifestyle, raises the risk of CVD by increasing the level of immobilization, and accelerates hyperlipidemia, high blood pressure, and diabetes. All these risk factors emphasize on the need for continuous monitoring and supporting weight management in nurses in order to prevent many of the subsequent physical complications. Comparing these results among shift workers and day workers suggests that the prevalence of overweight and obesity among the former group is significantly higher than the latter one. This conclusion confirms the results of previous studies; for example in a study by Griep a significant association between duration of exposure to night shift works and BMI was approved in nurses [32]. In a review conducted by Van Drongelen et al. the relationship between night shift work and body weight gain was affirmed. However they found that a crude association between shift work and body weight becomes insufficient after controlling for confounders [48]. Thus, in reaching a robust conclusion, several methodological limitations should be considered. Above all, in some of the studies weight and height data was gathered based on self-report techniques which might have brought about some ambiguities in BMI assessment. Additionally, most of the studies used crude data about the work schedule of the nursing staff at the moment of study, which could limit the conclusions regarding underlying relationships between study variables.

Based on study findings, the mean fasting blood sugar was estimated at 85.52 while 3.9% of study participants were at risk of diabetes due to fasting or random blood sugar upper limit of normal. Similarly, the relative risk was 3.8% in Gallagher's study but it was estimated at 1% in Zhao's study [22, 49]. However the reported prevalence of hyperglycemia among Bangladeshi nurses was significantly higher in Faruque study (19.4%) [21]. As it can be concluded from existing literature, mean fast blood sugar (FBS) in different studies varies from 71.84 in Hojat et al. research to 102.88 in Donker's study [50, 51]. Literature has affirmed that both low glucose level and impaired fasting glucose act as risk factors for coronary heart disease (fasting). In fact, CVD was regarded as the main cause of death in people living with diabetes, leading to 2/3 of mortalities in patients with type 2 diabetes [52].

Study results showed that, among nurses with shift works almost all risk factors got higher score representing the worse condition. Similar findings were observed in different studies conducted to compare the prevalence of FBS among shift workers and those working only day shifts as a nurse. It was concluded that shift workers encountered with an increased risk of

diabetes. Correspondingly, Pietroiusti's study found that 5% of night-shift workers suffered from high blood glucose, while only 3.9% of daytime workers experienced a similar condition [52]. Likewise, Hansen confirmed statistically significant increased risks of diabetes in nurses who worked night (1.58; 1.25 to 1.99) or evening shifts (1.29; 1.04 to 1.59) compared to those working in day shifts [30]. As the appropriateness of blood sugar level and controlling of inflammatory factors can play an important role in improving the platelet function and reduce the incidence of coronary syndromes it is recommended to develop preventive programs to control nurses' blood sugar and continuously monitor their well-being.

Analyzing nurses' lipid profiles in our study showed that their mean total cholesterol, LDL and HDL were 174.24, 91.54 and 48.1, respectively. Furthermore, 9.3% of nurses had hyper-cholesterolemia (TC≥200); 12.3% of them had LDL≥ 100 and 19.4% (CI 95%, 7.2–42.5) had HDL<40, which are associated with an increased risk of cardiovascular diseases. Literature affirmed that patients with increased cholesterol level were at higher risk of CVD. With high cholesterol, fatty deposits will develop in blood vessels, making it difficult for blood to flow through the arteries, causing a process called atherosclerosis, which potentially might lead to the rupture of clots and cause a heart attack or stroke [53]. Kazemi et al. affirmed that among different aspects of dyslipidemia (cholesterol >200mg/dl, HDL) low HDL (41.28%) and high LDL (35.54%) were the most prevalent. Low HDL and high TG are connected to increased levels of LDL, which consequently enhance the risk of having atherosclerosis. Furthermore, decreased cholesterol levels through statin medication or lifestyle intervention were associated with lower CVD risk [55]. Comparing the results based on work shifts total cholesterol as well as low-density lipoprotein (LDL) in shift workers were significantly higher compared to day workers [54]. Regarding the work schedule, the prevalence of fixed night-time shifts has been proven to be associated with sleep deficit, high cardiovascular risk, and association with coronary heart disease [55]. According to Solymanzadeh's study, a significant difference between total cholesterol and shift work was approved (p ≤0.001) and about 20% of shift workers were found to have hypercholesterolemia [17]. Furthermore, findings revealed no significant difference between high-density lipoprotein (HDL) and shift work (p = 0.77). In fact, only a small percentage (8.33%) of study population had low levels of HDL. Findings also revealed that night shift nurses had a significantly higher percentage of abnormal serum Triglyceride (59.3%) compared to day shift nurses (31.2%). In addition, the proportion of night shift nurses with abnormal LDL was significantly higher than the corresponding proportion among day shift nurses. These results emphasize that night shift nurses are at risk of abnormal lipid profile more than day workers [56]. On the contrary, no statistical significant relationship between lipid profile of nurses and their work shift was found (P = 0.051) [37]. This variation in obtained results might be due to different diets, amount and type of physical activity, and perhaps racial and ecological varieties in study populations. However, as a general approach standard preventive and treatment strategies such as diet control, aerobic exercise, and drug therapy are proposed to decrease serum cholesterol, alleviate plasma dyslipidemia and accordingly prevent heart disease among nurses.

To our knowledge, this is the first systematic review and meta-analysis conducted to investigate the prevalence of CVD risk factors among nurses reported in published studies between 2000 and 2022 using a wide range of search terms to retrieve all documents in English using strict criteria.

## Study limitations

One of the study limitations was that the study participants in our review included a sub-population of nurses not the general population. Second, our review has been limited to studies

published in English which potentially might exclude relevant studies. Third, there was lack of data for some of the countries which limited our capacity to increase the generalizability of findings to all geographical regions. Forth, our review mainly focused on investigating the CVD prevalence based on modifiable risk factors rather that age, gender or family history.

## Conclusion

As study results revealed, sedentary lifestyle was regarded as the main prevalent risk factor among nurses followed by family history of cardiovascular disease, being overweight and alcohol consumption. Furthermore, among nurses with shift works almost all risk factors got higher score representing the worse condition in comparison with day workers' nurses. In fact, the current study enables learning the associated risk factors of CVD among nurses to facilitate interventional programs with a view to reduce the exposure of nursing staff particularly those who work in shifts to cardiovascular risk factors. This information can comprise essential tools for health human resource management contributing to advance nursing.

## Supporting information

**S1 Checklist. PRISMA 2020 checklist.**
(DOCX)

**S1 File.**
(PDF)

## Author Contributions

**Conceptualization:** Saghar Khani, Ahmad Ghashghaee.

**Data curation:** Saghar Khani, Ahmad Ghashghaee.

**Formal analysis:** Ahmad Ghashghaee.

**Investigation:** Ahmad Ghashghaee.

**Methodology:** Sima Rafiei, Ahmad Ghashghaee.

**Project administration:** Sima Rafiei, Ahmad Ghashghaee.

**Resources:** Sima Rafiei.

**Software:** Sima Rafiei.

**Supervision:** Sima Rafiei, Ahmad Ghashghaee.

**Validation:** Sima Rafiei.

**Visualization:** Sima Rafiei, Ahmad Ghashghaee.

**Writing – original draft:** Saghar Khani, Ahmad Ghashghaee, Maryam Masoumi, Srva Rezaee, Golnaz Kheradkhah, Bahare Abdollahi.

**Writing – review & editing:** Saghar Khani, Ahmad Ghashghaee, Maryam Masoumi, Srva Rezaee, Golnaz Kheradkhah, Bahare Abdollahi.

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
