## [Decision Letter · Decision Letter 0]

17 Oct 2022

PONE-D-22-23328Cardiovascular risk factors among nurses: A global systematic review and meta-analysisPLOS ONE

Dear Dr. Ghashghaee,

Thank you for submitting your manuscript to PLOS ONE. After careful consideration, we feel that it has merit but does not fully meet PLOS ONE’s publication criteria as it currently stands. Therefore, we invite you to submit a revised version of the manuscript that addresses the points raised during the review process.

Dear, the systematic review with meta-analysis presents some methodological issues that need adjustments. Especially, about the publication of a protocol of systematic review and meta-analysis. Try registering with Prosper or OSF. Additionally, it has methodological and results-related comments that need to be rigorously corrected so that the article can be considered for publication.

We look forward to receiving your revised manuscript.

Kind regards,

Ana Claudia Morais Godoy Figueiredo, Ph.D

Academic Editor

PLOS ONE

Journal Requirements:

Reviewers' comments:

Reviewer's Responses to Questions

**Comments to the Author**

1. Is the manuscript technically sound, and do the data support the conclusions?

Reviewer #1: Yes

Reviewer #2: Yes

2. Has the statistical analysis been performed appropriately and rigorously? 

Reviewer #1: I Don't Know

Reviewer #2: Yes

3. Have the authors made all data underlying the findings in their manuscript fully available?

Reviewer #1: Yes

Reviewer #2: No

4. Is the manuscript presented in an intelligible fashion and written in standard English?

Reviewer #1: Yes

Reviewer #2: Yes

5. Review Comments to the Author

Reviewer #1: Some major and minor comments should be addressed before its publication.

The introduction is good. However, it is better for you to use more recent references in this field. For example, you can add the following two references to lines 4 and 11, respectively.

Shmakova NN, Puzin SN, Zarariy NS, Abol АV. THE CHARACTERISTICS OF THE IMPAIRED FUNCTIONS AND LIFE LIMITATIONS OF THE DISABLED PEOPLE DUE TO THE CORONARY HEART DISEASE. Journal of Population Therapeutics and Clinical Pharmacology. 2022 Mar 15;29(01).

Randhawa H, Ghaedi Y, Khan S, Al-Sharbatti S. The prevalence of overweight and obesity among health care providers in the emirate of Ajman, UAE. Journal of Complementary Medicine Research. 2020 Sep 25;11(3):40-.

It does not appear that a protocol has been pre-registered for this review (eg in PROSPERO). This is a concern as it introduces potential bias to the review and does not align with Cochrane's guidance. The authors should note that a protocol was not registered, and discuss it as a limitation in the discussion.

Pub Med? Please check the entire text of the manuscript in terms of writing

I suggest the author provide the full electronic search strategy for at least one database, including any limits used, such that it could be repeated, as a supplementary file.

Data screening and selection should have kappa statistics between the authors.

The authors can conduct a "snowball search" to add other articles (missing from this study).

I note the authors have reported results according to the PRISMA checklist. Please include a copy of the PRISMA checklist (with page numbers noted for each section) as a supplementary appendix, as required by PRISMA.

Results were duplicated in the text and tables/figures. I suggest avoiding any duplication to improve the readability of the manuscript.

I suggest an extensive revision of the discussion with more explanation of the results of the current study. In discussion, many factors could have influenced the results. Please also discuss other factors.

Reviewer #2: I am glad for the opportunity to review this manuscript. The theme of this systematic review is relevant.

I hope some questions and comments below may contribute to refining the paper.

Abstract

#1 – Objectives – In the last sentence, the authors should provide a more explicit statement of the study's primally objective (preferably with reference to PICOS or one of its variants)

#2 – Methods – The authors should specify the inclusion and exclusion criteria for the review and language restrictions.

#3 – Methods – The Newcastle-Ottawa Scale for quality assessment and the risk of bias assessment should be stated in the Methods.

#4 – Results – The results for family history of cardiovascular disease (41.9%; 95% IC: 29.8-55.1%), being overweight (33.3%; 95% IC: 24.7-43.2%), and alcohol consumption (24.6%; 95% IC: 16.4-35.2%) should be present since they are among the most prevalent risk factors in the study.

#5 – Result – Regarding systemic blood pressure, the authors should present the prevalence of systemic arterial hypertension instead of the mean systolic and diastolic blood pressures if they consider it a key study finding.

#6 - Results – The authors should provide a brief summary of the limitations of the evidence included in the review, such as the quality of the evidence and the risk of bias, inconsistency, and imprecision.

#7 – Conclusion – In addition to the general interpretation of the results and significant implications, the authors should present the main conclusions related to the study objective.

#8 – Did the authors previously register the study? If yes, the authors should provide the registered name and registration number in the Abstract and the main text.

Background

#1 – Are other systematic reviews addressing the cardiovascular risk factors among nurses? If yes, the authors should explain how the current systematic review can add to the acknowledgment from the previous studies.

#2 – In the last paragraph, the authors should provide a more explicit statement of the questions and objectives being addressed (preferably with reference to PICOS or one of its variants).

Methods

#1 – The literature review search period and the last search date should be stated in the first paragraph of the Methods.

#2 – The text presented in the 'Databases and search terms" should be split into separate headings, such as Study design and registration, Search strategies for identification of studies, Data collection and analysis, and Data extraction and management.

#3 – The authors should consider showing the complete search strategies for all databases, including any filters and limits used, in the Supplementary Material.

#4 – The authors should consider including a table summarizing the inclusion and exclusion criteria using the study characteristics (e.g., participants, setting, target condition(s), and study design) and report characteristics (e.g., years considered, language, and publication status).

#5 – The working definition for behavioral, biological, and physical risk factors should be presented in the Methods.

#6 – Why did the authors include only articles in English?

#7 – The study selection process should be described in more detail: the duplicate screening, title and abstract screening, and full-text articles assessment for relevance. Did a reference management software package or a screening and data extraction tool be used to screen for duplicates and study selection?

#8 – The risk of bias assessment should be stated in the Methods

#9 – The authors should indicate whether a review protocol exists and, if yes, the registered name, registration number, and where it can be accessed (e.g., Web address) in the Methods.

#10 – Figure 1 – The number of studies excluded for each criterion should be shown.

Results

#1 – In the text, before splitting into behavioral, biological, and physical risk factors, the authors could present the results of all cardiovascular risks together.

#2 – The authors should present the results of the meta-analysis as forest plots, including the heterogeneity test (I2).

#3 – Tables – Please correct the p-values that cannot be 0.000 (insert p < 0.001 instead p = 0.000).

#4 – Tables – The full form of abbreviations should be presented in the footnote.

#5 – Table 1 – The study's characteristics should include the study design/type.

Finally, I did not thoroughly check for grammatical errors since I am not a native English user. An appropriate language reviewer should do this.

6. PLOS authors have the option to publish the peer review history of their article (what does this mean?). If published, this will include your full peer review and any attached files.

Reviewer #1: No

Reviewer #2: No

---

## [Author Response · Author response to Decision Letter 0]

5 Jan 2023

Reviewer: 1 

Comment 1: The manuscript is technically sound and data support the conclusions.

Response to comment 1: We appreciate the positive attitude of the reviewer toward the manuscript.

Comment 2: The statistical analysis has been performed appropriately.

 Response to comment 2: We appreciate the positive attitude of the reviewer toward the statistical analysis. 

Comment 3: All data underlying the findings should be fully available in the supplementary file. 

Response to comment 3: We provided all data underlying the findings in the supplementary file. 

Comment 4: The manuscript is presented in an intelligible fashion.

Response to comment 4: We appreciate the positive attitude of the reviewer toward the manuscript presentation fashion.

Comment 5: The introduction is good but it is better to add some references in the field.

Response to comment 5: We added all suggested references in the introduction section.

Comment 6: The review protocol should have been pre-registered in PROSPERO. 

Response to comment 6: We pre-registered the review protocol and relevant information was added in to methodology section. This SLR has been registered on PROSPRO with code CRD42022329794

Comment 7: Provide the full electronic search strategies for databases in a supplementary file.

Response to comment 7: We added the required information in Method section.

Comment 8: Data screening and selection should have Kappa statistics between the authors.

Response to comment 8: We added Kappa statistics. 

Comment 9: Include a copy of PRISMA checklist as a supplementary appendix. 

Response to comment 9: We included a copy of PRISMA checklist in a supplementary appendix.

Comment 10: Results were duplicated in the text and tables/ figures.

Response to comment 10: We avoided any duplication to improve the readability of the manuscript. We only reported the main findings of the study in the text. 

Comment 11: More explanation of the results should be added to the discussion.

Response to comment 11: More explanation of results was added to the discussion.

Reviewer: 2 

Comment 1: Add objective to the end of introduction.

Response to comment 1: We added objective to the end of introduction.

Comment 2: Add inclusion and exclusion criteria for the review.

Response to comment 2: The inclusion and exclusion of the review are highlighted in green in the methodology section.

Comment 3: The Newcastle-Ottawa scale for quality assessment should be stated in the method.

Response to comment 3: The Newcastle-Ottawa scale for quality assessment was explained in the method and highlighted in green.

Comment 4: The results for family history of cardiovascular disease, being overweight and alcohol consumption should be present since they are among the most prevalent risk factors in the study

Response to comment 4: We added the mentioned factors as most prevalent risk factors in the study.

Comment 5: Regarding systemic blood pressure, the authors should present the prevalence of systemic arterial hypertension instead of the mean systolic and diastolic blood pressures.

Response to comment 5: This was one of the goals of our research, but, unfortunately these type of data did not exist on the included articles 

Comment 6: Provide a summary of limitations of the evidence included in the review.

Response to comment 6: We provided a summary of limitations at the end of discussion section.

Comment 7: Present the main conclusions related to the study objective.

Response to comment 7: The main conclusions were presented regarding the study objective.

Comment 8: How the current systematic review can add to the acknowledgment from the previous studies.

Response to comment 8: We highlighted the explanation about what this systematic review adds to previous studies (in the last paragraph of introduction).

Comment 9: In the last paragraph, the authors should provide a more explicit statement of the objectives being addressed.

Response to comment 9: We stated the study objective in the last paragraph of introduction.

Comment 10: The literature review search period and the last search date should be stated in the first paragraph of the Methods.

Response to comment 10: The required information (The search was carried out from the beginning of the year 2000 to the end of March 2022) is highlighted in the methodology. 

Comment 11: Data bases and search terms should be splited in to separate headings of study design and registration, search strategies, data collection and analysis, and data extraction and management. 

Response to comment 11: The methodology section was splited based on the reviewer’s comment. 

Comment 12: The authors should consider showing the complete search strategies for all databases, including any filters and limits used, in the Supplementary Material.

Response to comment 12: We provided the required data in the Method section.

Comment 13: The authors should consider the inclusion and exclusion criteria using the study characteristics and report characteristics.

Response to comment 13: We considered the inclusion and exclusion criteria in the methodology section as highlighted in green.

Comment 14: The working definition for behavioral, biological, and physical risk factors should be presented in the Methods.

Response to comment 14: As we searched for cardiovascular risk factors as a whole search term, we did not divide our searching process in to behavioral, biological and physical risk factors. Thus, in the result section we reported all identified risk factors all together irrespective of their category. 

Comment 15: Why did the authors include only articles in English?

Response to comment 15: We only searched for articles in English and we regarded this as a study limitation. 

Comment 16: The study selection process should be described in more detail: the duplicate screening, title and abstract screening, and full-text articles assessment for relevance. Did a reference management software package or a screening and data extraction tool be used to screen for duplicates and study selection?

Response to comment 16: The title/ abstract screening, full-text assessment, and a reference management software package used to screen for duplicates were mentioned in the methodology section.

Comment 17: The risk of bias assessment should be stated in the Methods. 

Response to comment 17: The risk of bias assessment was stated in the methodology section (identified by NOS tool). 

Comment 18: The number of studies excluded for each criterion should be shown in Figure 1.

Response to comment 18: The number of studies excluded for each criterion was shown in Figure 1.

Comment 19: In the text, before splitting into behavioral, biological, and physical risk factors, the authors could present the results of all cardiovascular risks together.

Response to comment 19: We presented the results of all cardiovascular risks together as presented in Table 2. 

Comment 20: The authors should present the results of the meta-analysis as forest plots, including the heterogeneity test (I2).

Response to comment 20: Based on the data and type of analysis, Table was preffered rather than forest plot.

Comment 21: Please correct the p-values that cannot be 0.000 (insert p < 0.001 instead p = 0.000).

Response to comment 21: We made necessary corrections.

Comment 22: The full form of abbreviations should be presented in the footnote.

Response to comment 22: We made necessary changes.

---

## [Decision Letter · Decision Letter 1]

8 May 2023

PONE-D-22-23328R1Cardiovascular risk factors among nurses: A global systematic review and meta-analysisPLOS ONE

Dear Dr. Ghashghaee,

Thank you for submitting your manuscript to PLOS ONE. After careful consideration, we feel that it has merit but does not fully meet PLOS ONE’s publication criteria as it currently stands. Therefore, we invite you to submit a revised version of the manuscript that addresses the points raised during the review process.

We look forward to receiving your revised manuscript.

Kind regards,

Patricia Khashayar

Academic Editor

PLOS ONE

Journal Requirements:

Reviewers' comments:

Reviewer's Responses to Questions

**Comments to the Author**

1. If the authors have adequately addressed your comments raised in a previous round of review and you feel that this manuscript is now acceptable for publication, you may indicate that here to bypass the “Comments to the Author” section, enter your conflict of interest statement in the “Confidential to Editor” section, and submit your "Accept" recommendation.

Reviewer #1: All comments have been addressed

Reviewer #2: All comments have been addressed

2. Is the manuscript technically sound, and do the data support the conclusions?

Reviewer #1: Yes

Reviewer #2: Yes

3. Has the statistical analysis been performed appropriately and rigorously? 

Reviewer #1: Yes

Reviewer #2: Yes

4. Have the authors made all data underlying the findings in their manuscript fully available?

Reviewer #1: Yes

Reviewer #2: Yes

5. Is the manuscript presented in an intelligible fashion and written in standard English?

Reviewer #1: Yes

Reviewer #2: Yes

6. Review Comments to the Author

Reviewer #1: I thank the authors for the sufficient answers to the previous comments. In my opinion, the study is clear and can be suitable for publication

Reviewer #2: Thank you for responding to the comments from all reviewers and resubmitting your work. The authors have made appropriate adjustments to the original submission, and the manuscript has improved from the feedback from the reviewers. I have only a few comments/suggestions regarding the Abstract Section to enhance your manuscript further:

The key inclusion and exclusion criteria for the review and language restrictions should be stated in the Abstract Methods.

The Newcastle-Ottawa Scale for quality assessment and the risk of bias assessment should be stated in the Abstract Method.

Regarding the risk factors, family history of cardiovascular disease (41.9%; 95% IC: 29.8-55.1%), being overweight (33.3%; 95% IC: 24.7-43.2%), and alcohol consumption (24.6%; 95% IC: 16.4-35.2%) should be present in the Abstract Results since they are among the most prevalent in the study.

The authors should summarize the limitations of the evidence included in the review, such as the quality of the evidence and the risk of bias, inconsistency, and imprecision in the Abstract Results.

7. PLOS authors have the option to publish the peer review history of their article (what does this mean?). If published, this will include your full peer review and any attached files.

Reviewer #1: No

Reviewer #2: No

---

## [Author Response · Author response to Decision Letter 1]

11 May 2023

Comment 1: The key inclusion and exclusion criteria for the review and language restrictions should be stated in the Abstract Methods.

Response to comment 1: from the beginning of 2000 to March 2022

Comment 2: The Newcastle-Ottawa Scale for quality assessment and the risk of bias assessment should be stated in the Abstract Method.

 Response to comment 2: the Newcastle-Ottawa Scale (NOS) was used to assess the methodological quality of included studies.

Comment 3: Regarding the risk factors, family history of cardiovascular disease (41.9%; 95% IC: 29.8-55.1%), being overweight (33.3%; 95% IC: 24.7-43.2%), and alcohol consumption (24.6%; 95% IC: 16.4-35.2%) should be present in the Abstract Results since they are among the most prevalent in the study.

Response to comment 3: Also family history of cardiovascular disease (41.9%; 95% IC: 29.8-55.1%), being overweight (33.3%; 95% IC: 24.7-43.2%), and alcohol consumption (24.6%; 95% IC: 16.4-35.2%) was found among the participants.

Comment 4: The authors should summarize the limitations of the evidence included in the review, such as the quality of the evidence and the risk of bias, inconsistency, and imprecision in the Abstract Results.

Response to comment 4: By adding this section to the abstract, it would exceeded the limitation word.

---

## [Editor Report · Decision Letter 2]

12 May 2023

Cardiovascular risk factors among nurses: A global systematic review and meta-analysis

PONE-D-22-23328R2

Dear Dr. Ghashghaee,

We’re pleased to inform you that your manuscript has been judged scientifically suitable for publication and will be formally accepted for publication once it meets all outstanding technical requirements.

Kind regards,

Patricia Khashayar

Academic Editor

PLOS ONE
---

## [Editor Report · Acceptance letter]

1 Dec 2023

PONE-D-22-23328R2 

Cardiovascular risk factors among nurses: A global systematic review and meta-analysis 

Dear Dr. Ghashghaee:

I'm pleased to inform you that your manuscript has been deemed suitable for publication in PLOS ONE. Congratulations! Your manuscript is now with our production department. 

Kind regards, 

on behalf of

Dr. Patricia Khashayar 

Academic Editor

PLOS ONE